# Comparison of Prognostic Performance between Neuron-Specific Enolase and S100 Calcium-Binding Protein B Obtained from the Cerebrospinal Fluid of Out-of-Hospital Cardiac Arrest Survivors Who Underwent Targeted Temperature Management

**DOI:** 10.3390/jcm10071531

**Published:** 2021-04-06

**Authors:** Changshin Kang, Wonjoon Jeong, Jung Soo Park, Yeonho You, Jin Hong Min, Yong Chul Cho, Hong Joon Ahn

**Affiliations:** 1Department of Emergency Medicine, Chungnam National University Hospital, 282, Munhwa-ro, Jung-gu, Daejeon 35015, Korea; changsiny@naver.com (C.K.); gardenjun@hanmail.net (W.J.); yyh1003@hanmail.net (Y.Y.); boxter73@naver.com (Y.C.C.); jooniahn@hanmail.net (H.J.A.); 2Department of Emergency Medicine, College of Medicine, Chungnam National University, Daejeon 35015, Korea; shiphid@hanmail.net; 3Department of Emergency Medicine, Chungnam National University Sejong Hospital, 20, Bodeum 7-ro, Sejong 30099, Korea

**Keywords:** out-of-hospital cardiac arrest, targeted temperature management, prognosis, biomarkers, phosphopyruvate hydratase, neuron-specific enolase, S100 proteins

## Abstract

We compared the prognostic performances of serum neuron-specific enolase (sNSE), cerebrospinal fluid (CSF) NSE (cNSE), and CSF S100 calcium-binding protein B (cS100B) in out-of-hospital cardiac arrest (OHCA) survivors. This prospective observational study enrolled 45 patients. All samples were obtained immediately and at 24 h intervals until 72 h after the return of spontaneous circulation. The inter- and intragroup differences in biomarker levels, categorized by 3 month neurological outcome, were analyzed. The prognostic performances were evaluated with receiver operating characteristic curves. Twenty-two patients (48.9%) showed poor outcome. At all-time points, sNSE, cNSE, and cS100B were significantly higher in the poor outcome group than in the good outcome group. cNSE and cS100B significantly increased over time (baseline vs. 24, 48, and 72 h) in the poor outcome group than in the good outcome group. sNSE at 24, 48, and 72 h showed significantly lower sensitivity than cNSE or cS100B. The sensitivities associated with 0 false-positive rate (FPR) for cNSE and cS100B were 66.6% vs. 45.5% at baseline, 80.0% vs. 80.0% at 24 h, 84.2% vs. 94.7% at 48 h, and 88.2% (FPR, 5.0%) vs. 94.1% at 72 h. High cNSE and cS100B are strong predictors of poor neurological outcome in OHCA survivors. Multicenter prospective studies may determine the generalizability of these results.

## 1. Introduction

Despite numerous medical studies and clinical efforts to improve the prognosis of cardiac arrest (CA) survivors with successful return of spontaneous circulation (ROSC), achieving competent mortality and neurological outcomes in this patient population remains challenging [1]. In a previously published report, only 8% of patients showed good neurological outcomes after receiving post-cardiac arrest service for out-of-hospital cardiac arrest (OHCA) [2]. Since most comatose patients with poor neurological outcomes are considered eligible for withdrawal of life-sustaining treatment (WLST), accurate neurological prognostication is of utmost importance to avoid futile treatments in patients destined for poor outcome and inappropriate WLST in patients who may have a chance of neurological recovery [1].

For these reasons, many studies have attempted to develop accurate and efficient neurological prognostication in CA survivors. In particular, various biomarkers have been investigated because of their advantages over other tools, such as the ability to obtain quantitative results and the likely independence from the effects of sedatives [3]. A number of studies reported that neuron-specific enolase (NSE) and S100 calcium-binding protein B (S100B), which are released due to brain injury, are associated with neurological outcome in OHCA survivors [4,5,6,7]. However, the accuracy and efficiency of serum values remain debatable because the serum samples obtained for measurement of these biomarkers are often contaminated [1,8]. Thus, cerebrospinal fluid (CSF) samples have been suggested as an alternative to reduce contamination, despite the difficulty in obtaining them [1]. Our previously published study showed that, in comparison with serum NSE evaluations, assessments of CSF NSE concentrations could yield earlier and more accurate prognostication in OHCA survivors treated with targeted temperature management (TTM) [9].

S100B, which shows biochemically different characteristics from NSE, such as a lower molecular weight (10.7–21 vs. 78 kDa) and shorter half-life (2.0 vs. 24 h), has also been reported to be a useful additional biomarker for neurological prognostication in OHCA survivors [10,11,12]. However, to the best of our knowledge, CSF S100B has not been assessed as a prognostic biomarker in OHCA survivors. Therefore, this study aimed to assess the usefulness of CSF S100B as a prognostic biomarker and to compare the neurological prognostic performance of CSF NSE and S100B in OHCA survivors. In addition, the prognostic performance of serum NSE, which is a recommended prognostic biomarker in CA survivors according to current international guideline [13], was analyzed at the same time points to clearly compare the prognostic values of these CSF biomarkers.

## 2. Patients and Methods

### 2.1. Study Design and Patients

The present study was performed prospectively on adult comatose OHCA survivors treated with TTM at Chungnam National University Hospital (CNUH) in Daejeon between February 2019 and August 2020. This study was approved by the CNUH Institutional Review Board (CNUH-2017-10-027), and informed consent was obtained from the patients’ designated decision-makers prior to enrolment. Patients were subsequently excluded if the patient’s next of kin declined provision of further treatment after CA or if the patients (1) were younger than 18 years, (2) had experienced CA due to trauma, (3) failed to maintain a temperature of 33 °C for TTM due to unstable hemodynamics, (4) were ineligible for lumbar catheter placement (e.g., no consent from the patient’s family; needed to maintain anticoagulation or antiplatelet therapy after early percutaneous coronary intervention or extracorporeal membrane oxygenation; or showed severe brain edema in imaging studies, such as obliteration of the basal cisterns, pseudo-subarachnoid hemorrhage, and visible intracranial mass), (6) had a medical history of cerebral disease (e.g., dementia, cerebral infarction, cerebral hemorrhage, parkinsonism, or brain tumor).

### 2.2. Target Temperature Management Protocol

Patients were managed according to our previously published TTM protocol [14]. In this protocol, TTM was induced using ice packs; intravenous cold saline; and TTM devices, namely, Arctic Sun^®^ and Energy Transfer Pads™ (Bard Medical, Louisville, CO, USA). A target temperature of 33 °C was maintained for 24 h and monitored with a bladder probe. Upon completion of the TTM-maintenance phase, the patients were rewarmed at a rate of 0.25 °C per hour to 37 °C. Midazolam (0.05 mg/kg intravenous bolus, followed by a titrated intravenous continuous infusion of 0.05–0.2 mg/kg/h) and cisatracurium (0.15 mg/kg intravenous bolus, followed by an infusion of up to 0.3 mg/kg/h) were administered for sedation and to control shivering. All patients received standard intensive care according to our institutional intensive care unit protocol.

### 2.3. Measurement of NSE and S100B Values Obtained from Both Serum and Cerebrospinal Fluid

All serum and CSF samples were repeatedly obtained, immediately and at 24 h intervals until 72 h (sNSE_i,24,48,72_, cNSE_i,24,48,72_, and cS100B_i,24,48,72)_ after ROSC via venipuncture and lumbar catheter drainage, respectively. The procedure for lumbar catheterization was performed with the patient lying in the lateral decubitus position, with the neck, hips, and knees flexed. The lumbar catheter was inserted using a Hermetic TM lumbar accessory kit (Integra Neurosciences, Plainsboro, NJ, USA) under aseptically guided sonography by an expert physician. All samples were analyzed at one laboratory, namely Green Cross Laboratory (GC Labs; Yongin, Geonggi-do, Korea). NSE and S100B concentrations were determined using an electrochemiluminescence immunoassay (ECLIA) with Elecsys NSE^®^ (COBAS e801; Roche Diagnostics, Rotkreuz, Switzerland) and Elecsys S100^®^ (COBAS e411; Roche Diagnostics, Mannheim, Germany), respectively. The measurement ranges of NSE and S100B were 0.1–300 ng/mL and 0.005–30 μg/L, respectively. GC Labs was compliant with the relevant national and international guidelines.

### 2.4. Outcomes and Data Collection

The primary outcome in this study was poor neurological outcome at 3 months after ROSC. Neurological outcomes were measured using the Glasgow–Pittsburgh cerebral performance category (CPC) scale, either through face-to-face interviews or structured telephone interviews [15]. The interviews were conducted by an emergency physician who was well-versed with our protocols and was blinded to the patients’ prognosis and clinical data. The patients were classified into five categories on the basis of CPC scores: CPC 1 (good performance), CPC 2 (moderate disability), CPC 3 (severe disability), CPC 4 (vegetative state), and CPC 5 (brain death or death).

The following baseline demographics and characteristics of the enrolled patients were recorded: age, sex, presence of a witness on collapse, bystander cardiopulmonary resuscitation (CPR), first monitored cardiac rhythm, etiology of CA, no- and low-flow time from CA, Glasgow coma score (GCS) immediately after ROSC, and time from ROSC to lumbar puncture.

### 2.5. Statistical Analysis

The categorical variables are presented as frequencies and percentages. They were compared using Fisher’s exact tests. Continuous variables were presented as means with standard deviations or medians with the interquartile range (IQR) depending on the normality of the data. Differences in the concentrations of biomarkers obtained from the serum or CSF between good and poor neurological outcomes were assessed using Student’s t-test or Mann–Whitney U-test. Repeated-measures ANOVA or Friedman test with Wilcoxon signed rank-sum test after Bonferroni correction was used to evaluate the changes in serum and CSF NSE and S100B values over time in each neurological outcome group.

To assess the predictive accuracy for a 3 month poor neurological outcome, receiver operating characteristic (ROC) curves were constructed for the prognostic values of NSE and S100B. The ROC curve plots the sensitivity of a measure on the *y*-axis and (1-sensitivity) on the *x*-axis, and it measures the overall accuracy of a test. The most important summary index of the ROC curve is the area under the ROC curve (AUROC). As high specificity is the most critical metric for use in CA, in which false-positive predictions may lead to the death of the patient, we set the cutoff values to at least 95% specificity based on the Touden index. The DeLong method was used to perform a pairwise comparison of the AUROC curves of NSE and S100B (serum NSE vs. CSF NSE, serum NSE vs. CSF S100B, and CSF NSE vs. CSF S100B) concentrations at each time point [16].

Statistical analyses were performed using IBM SPSS Statistics version 25.0 (IBM Corp., Armonk, NY, USA) and MedCalc version 15.2.2 (MedCalc Software, Mariakerke, Belgium). The results were considered significant at *p* < 0.05.

## 3. Results

### 3.1. Characteristics of the Study Population

A total of 60 comatose OHCA patients were treated with TTM during the study period. Among them, 15 patients were excluded for the following reasons: 10 were ineligible for lumbar catheter placement, the kin of one patient did not provide consent for the study, and 4 had a known medical history of cerebral disease. Thus, 45 patients were included in this study, of which 22 (48.9%) showed poor neurological outcome, corresponding to a primary outcome defined by a CPC score of 3 to 5 (Figure 1). The basic characteristics of the enrolled patients are given in Table 1. The group with poor neurological outcomes showed significantly lower instances of witnesses during CA, bystander CPR, cardiac etiology, shockable rhythm, and GCS and significantly longer no- and low-flow times (Table 1).

ROSC, return of spontaneous circulation; GCS, Glasgow coma score; TTM, targeted temperature management; ECMO, extracorporeal membrane oxygenation; PCI, percutaneous coronary intervention.

### 3.2. Comparison of NSE and S100B Concentrations between Groups with Good and Poor Outcomes

All the serum and CSF NSE values and CSF S100B values obtained in the poor neurological outcome group were significantly higher than the values obtained at the same time points in the good neurological outcome group (Table 2 and Figure 2)

Figure 2 shows the change in serum and CSF NSE and CSF S100B concentrations according to the neurologic outcome over time from ROSC. The Friedman test with post hoc analysis showed that, compared to the serum NSE_i_, serum NSE_24_ and NSE_48_ were significantly higher in the poor neurological outcome group (serum NSE_24_ and NSE_48,_
*p* < 0.005 and 0.007; Figure 2), while in the good neurological outcome group, the serum NSEs measured at 24, 48, and 72 h were not significantly different compared to NSE_i_ (*p* > 0.008; Figure 2). The CSF NSE and S100B concentrations at 24, 48, and 72 h after ROSC were significantly higher than those measured immediately after ROSC in the group with a poor neurological outcome (CSF NSE_24_, NSE_48_, and NSE_72_, *p* < 0.001, < 0.001, and 0.002; CSF S100B_24_, S100B_48_, and S100B_72_, *p* = 0.002, 0.001, and 0.002; Figure 2), while in the group showing a good neurological outcome, the CSF NSE_24_, NSE48, and S100B_72_ concentrations were significantly higher and lower, respectively, than the CSF NSE_i_ and S100B_i_ (CSF NSE_24_ and NSE_48_, *p* = 0.002 and < 0.001; S100B_72_, *p* = 0.004; Figure 2).

### 3.3. Receiver Operating Characteristic Analysis for Predicting Poor Neurological Outcomes

Figure 3 shows the AUROC values of NSE and S100B concentrations for poor neurological outcome at each time point, from immediately to 72 h after ROSC. At 24 and 48 h after ROSC, both CSF NSE and S100B showed significantly higher AUROC values than serum NSE and CSF S100B measured at 72 h from ROSC had significantly higher prognostic performance for poor neurological outcome compared with serum NSE measured at the same time point. However, both CSF biomarkers measured immediately after ROSC and CSF NSE measured at 72 h showed relatively higher AUROC values compared with serum NSE, though it was not statistically significant (Figure 3). Even though statistical comparison of the AUROC values between CSF NSE and S100B were not significant at all time points, the AUROC values of CSF S100B were relatively higher or similar to that of CSF NSE (Figure 3).

Table 3 shows the cutoff concentrations showing the highest specificity (at least 95%) and the corresponding sensitivities of serum NSE and CSF biomarkers. Serum NSE at all time points showed lower sensitivity than CSF NSE and S100B (Table 3). In a comparison between CSF NSE and S100B, CSF S100B had higher sensitivities at 48 and 72 h from ROSC than CSF NSE (84.2% [95% CI, 60.4–96.6] vs. 94.7% [95% CI, 74.0–99.9] at 48 h, 88.2% [95% CI, 63.6–98.5] with 95% specificity vs. 94.1% [95% CI, 71.3–99.9] at 72 h; Table 3)

## 4. Discussion

This study analyzed serum NSE, CSF NSE, and S100B concentrations for the first 72 h after ROSC in OHCA survivors who underwent TTM and compared their prognostic performance for a poor neurological outcome 3 months after ROSC. CSF NSE and S100B showed significantly higher prognostic performances compared with serum NSE at all time points, except soon after ROSC. The most powerful prognostic performances of CSF NSE and S100B, with the highest sensitivity and 0 to 5% false-positive rate (FPR) for poor neurological outcome, were analogously observed at 48 h after ROSC in this study. The predominance of AUROC values between NSE and S100B fluctuated over time from ROSC until 72 h. Although the differences in AUROC values between NSE and S100B measurements at all four time points did not achieve statistical significance, S100B showed relatively higher AUROC values and sensitivities than NSE for poor neurological outcome immediately and at 48 and 72 h, and at 48 and 72 h from ROSC, respectively. These findings may indicate that CSF NSE and S100B concentrations are independently increased by their own specific pathophysiologic mechanisms according to the treatment course during TTM in OHCA survivors.

Since the CSF sample has the advantage that the biomarker need not be transported across the blood–brain barrier (BBB) for detection, thus greatly reducing contamination, the CSF sample for biomarkers derived from damaged cells in the central nervous system was suggested [1]. In the 1990s, Kärkelä et al. reported that the CSF NSE measured at 76 h from resuscitation was the most distinct between the groups the neurologically recovered and the disabled patient; however, the prognostic performances of CSF NSE such as AUROC value, sensitivity, or specificity were not demonstrated due to insufficient sample size (*n* = 20) [17]. Recently, we reported the usefulness of the CSF concentration of NSE in predicting neurological outcome in OHCA survivors [9]. Through the present study, we not only emphasize the usefulness of measuring CSF biomarker concentration to predict neurological outcome in OHCA survivors but also suggest that CSF S100B can be a useful prognostic biomarker in OHCA survivors.

In this study, the CSF S100B concentration measured immediately after ROSC showed a relatively higher AUROC value than the CSF NSE concentration at the same time point (AUROC, 0.95 vs. 0.90). This result is consistent with the findings of previous studies [10,11]. Bottiger et al. [10] reported that significant differences in serum S100B concentrations between patients showing good and poor neurological outcomes were apparent 30 min after initiation of CPR, a time point much earlier than the 24 h at which significant differences in serum NSE concentrations appeared. In addition, Zellner et al. [11] reported that the serum S100B concentration showed a significant difference between patients with good and poor neurological outcomes from admission to day 1 post-ROSC whereas the corresponding changes in the NSE concentration appeared from day 1 to day 2 post-ROSC. In previous studies comparing the serum values and prognostic performance between NSE and S100B, the difference in molecular characteristics between the two biomarkers (e.g., half-lives and molecular weights) was suggested as a hypothetical explanation for the increasing serum values and superior prognostic performance of S100B at an earlier stage when compared with NSE [10,11]. In several previous studies, the molecular characteristics of S100B were assumed to enable its easy infiltration into the serum across the impaired BBB and its rapid washout in the serum [4,11,18].

Direct sampling of NSE and S100B from the CSF can eliminate interferences related to half-life and molecular weight, both of which influence the detection and clearance time in serum and the ability to penetrate the BBB, and was performed in this study. However, interestingly, the results of this study were similar to those of previous studies in which the initial serum S100B concentration showed a relatively higher prognostic performance than the serum NSE concentration. These findings suggest that, while molecular characteristics such as molecular weight and half-life, are important, they could not fully explain the variations in the values of NSE and S100B obtained from the serum or CSF. Therefore, we suggest some possible explanations below.

S100B is dominantly expressed in astrocytes and glial cells of the grey matter [19,20], which is more severely and readily affected than the white matter during hypoxic-ischemic brain injury after CA [21]. Based on the pathophysiology of a brain injury after CA, several studies have reported that a decreased grey to white matter ratio (GWR) is associated with neurological outcome in OHCA survivors [22,23,24,25], and they further explained that the decreased GWR may be caused by cytotoxic edema in the grey matter, represented as hypoattenuation in brain computed tomography rather than pathophysiologic changes in the white matter [22]. Moreover, Hol and Pekny have suggested that an early increase in glial markers such as S100B is a result of early glial cell activation, since glial activation is a common element in brain trauma and ischemia [26]. Therefore, we suggest that the greater vulnerability of grey matter to hypoxic damage could account for the earlier S100B release into the serum and CSF; thus, this could explain the relatively higher AUROC value of CSF S100B immediately after ROSC in comparison with the value of CSF NSE. Hence, S100B derived from the grey matter of the brain may be more useful than NSE as a prognostic biomarker in OHCA survivors within approximately 19 h from ROSC, since our maximal lumbar puncture time in this study was 19.5 h after ROSC.

In the good neurological outcome group, the repeated paired statistical comparisons of CSF NSE and S100B concentrations between immediately and 24 h after ROSC showed ascending and parallel patterns. Considering the discrepancy in half-lives between NSE and S100B, the trends in these biomarkers between immediately and 24 h after ROSC may be interpreted as continuous excretion into the CSF space resulting from ongoing cerebral damage in OHCA survivors with good neurological outcomes. However, in two other previous studies using serum analysis, NSE and S100B concentrations showed parallel and descending patterns between baseline and at 24 h from ROSC in the good neurological outcome group [6,27], in contrast to our study. These non-decreasing CSF levels (i.e., indicating continuous excretion) of biomarkers representing cerebral damage from immediately to 24 h from ROSC might be presumed to represent secondary cerebral damage in the “two-hit” model, which occurred in the hours and days following CA and reperfusion [28]. Therefore, it was more clearly showed through our CSF analysis that ischemic-reperfusion cerebral injury following ROSC occurred in even the good neurological outcome group after CA, which was not observed in previous serum analysis studies.

These findings have several limitations with regard to generalization. First, this study involved an insufficient number of patients. A total of 15 patients who underwent TTM were excluded due to their ineligibility for lumbar catheter placement and a known medical history of cerebral disease. This could have caused selection bias and limited the generalizability of these results. However, in six patients, the values of biomarkers derived from the brain were modified by a known medical history of cerebral disease and severe brain edema (Figure 1). Further studies are required to reduce biases from sample numbers. Second, since the range of measured CSF NSE and S100B values was based on serum samples, the upper limits of the measurement ranges of NSE and S100B were 300 ng/mL and 30 μg/L, respectively. The CSF samples with measured values above the upper limit were not analyzed further with dilution. In particular, the NSE concentration measured 72 h after ROSC showed the maximal value even in the good neurological outcome group. However, the normal range of CSF values according to age, sex, and comorbidities is still unclear. Thus, clear definitions of normal ranges need to be established in further studies that report on CSF biomarkers. Finally, in some cases, we could not fully obtain the CSF samples until 72 h because of death prior to complete TTM or non-function of the lumbar catheter.

## 5. Conclusions

High CSF NSE and S100B concentrations are strong predictive markers of poor neurological outcome in OHCA survivors, even immediately after ROSC. In particular, CSF S100B concentration showed higher sensitivities with 0% FPR at 48 and 72 h after ROSC than with CSF NSE. Further multicenter studies with large sample sizes are required to generalize our results.

## Figures and Tables

**Figure 1 jcm-10-01531-f001:**
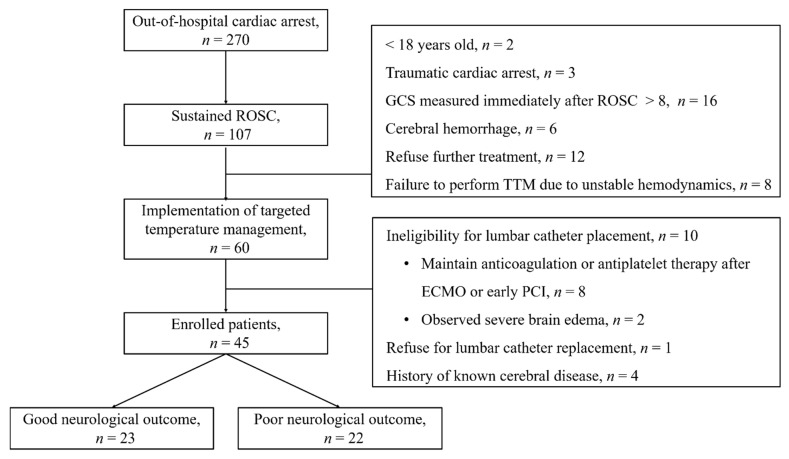
Flow diagram of patients included in this study.

**Figure 2 jcm-10-01531-f002:**
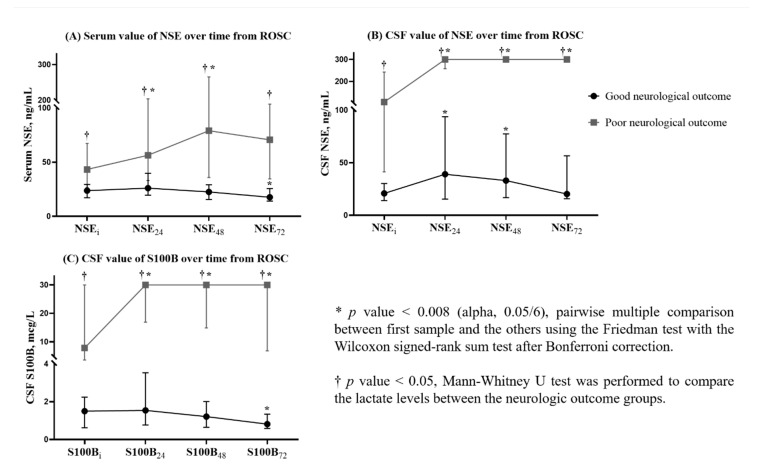
Changes in the concentrations of (**A**) serum NSE, (**B**) CSF NSE, and (**C**) CSF S100B measured immediately and at 24, 48, and 72 h after ROSC, categorized according to neurological outcomes. The line and the plot indicate the median levels and interquartile range of biomarkers. The concentrations of all biomarkers measured at all time points were significantly higher in those with poor neurological outcome than in those with good neurological outcome. CSF, cerebrospinal fluid; ROSC, return of spontaneous circulation; NSE, neuron-specific enolase concentration measured immediately (NSE_i_) and at 24 (NSE_24_), 48 (NSE_48_), and 72 h (NSE_72_) after the return of spontaneous circulation; S100B, S100B protein concentration measured immediately (S100B_i_) and at 24 (S100B_24_), 48 (S100B_48_), and 72 h (S100B_72_) after the return of spontaneous circulation.

**Figure 3 jcm-10-01531-f003:**
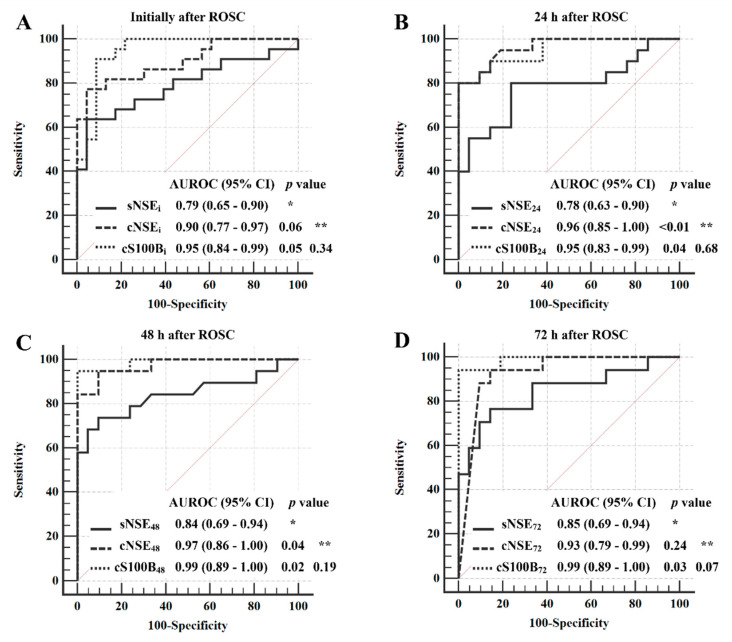
Statistical analysis of the area under the receiver operating characteristic curves of serum and cerebrospinal fluid neuron-specific enolase, and cerebrospinal fluid S100B concentrations measured (**A**) immediately and at (**B**) 24, (**C**) 48, and (**D**) 72 h after the return of spontaneous circulation. References for the DeLong test for statistical comparison between (*) serum NSE and CSF biomarkers and (**) CSF NSE and S100B. CI, confidence interval; NSE, serum and cerebrospinal fluid neuron-specific enolase concentration measured immediately (sNSE_i_ and cNSE_i_) and at 24 (sNSE_24_ and cNSE_24_), 48 (sNSE_48_ and cNSE_48_), and 72 h (sNSE_72_ and cNSE_72_) after the return of spontaneous circulation; S100B, cerebrospinal fluid S100B protein concentration measured immediately (cS100B_i_) and at 24 (cS100B_24_), 48 (cS100B_48_), and 72 h (cS100B_72_) after the return of spontaneous circulation.

**Table 1 jcm-10-01531-t001:** Baseline demographics and characteristics of 44 patients.

	Total Patients (*n* = 45)	Good Neurological Outcome (*n* = 23)	Poor Neurological Outcome (*n* = 22)	*p*
Age, years, median (IQR)	57 (18–87)	58 (18–81)	51 (19–87)	0.73
Male, n (%)	36 (80.0)	21 (91.3)	15 (68.2)	0.07
Cardiac arrest characteristics				
Witness, n (%)	28 (62.2)	18 (78.3)	10 (45.5)	0.03
Bystander CPR, n (%)	30 (66.7)	19 (82.6)	11 (50.0)	0.03
Shockable rhythm, n (%)	14 (31.1)	14 (60.9)	0	<0.01
Cardiac etiology, n (%)	20 (44.4)	15 (65.2)	5 (22.7)	0.02
No flow time, min, median (IQR)	4.0 (0.0–90.0)	1.0 (0.0–30.0)	12.0 (0.0–90.0)	<0.01
Low flow time, min, median (IQR)	18.5 (2.0–58.0)	12.0 (2.0–40.0)	26.0 (2.0–58.0)	<0.01
GCS immediately after ROSC	3 (3–8)	3 (3–8)	3 (3–3)	<0.01
Time to LP from ROSC, hour, median (IQR)	4.5 (1.2–19.5)	3.4 (1.2–11.6)	5.0 (2.3–19.5)	0.11

IQR, interquartile range; CPR, cardiopulmonary resuscitation; GCS, Glasgow coma scale; ROSC, restoration of spontaneous circulation; CSF, cerebrospinal fluid; LP, lumbar puncture.

**Table 2 jcm-10-01531-t002:** The serum and cerebrospinal fluid concentrations of neuron-specific enolase (NSE) and S100 calcium-binding protein B (S100B) in good and poor neurological outcome groups.

Value	Total Patients (*n* = 45)	Good Neurological Outcome (*n* = 23)	Poor Neurological Outcome (*n* = 22)	*p* Value
Serum NSE, ng/mL, median (IQR)				
NSE_i_,	29.0 (13.8–208.0)	23.8 (14.1–49.3)	43.4 (13.8–208.0)	<0.01
NSE_24_	33.7 (12.8–300.0)	26.1 (12.8–75.1)	56.5 (16.6–300.0)	<0.01
NSE_48_	29.2 (9.3–300.0	22.6 (9.3–54.6)	79.1 (11.5–300.0)	<0.01
NSE_72_	25.7 (8.0–300.0)	17.7 (8.0–65.3)	70.7 (12.4–300.0)	<0.01
CSF NSE, ng/mL, median (IQR)				
NSE_i_,	30.8 (8.0–300.0)	20.9 (8.0–60.3)	106.8 (17.4–300.0)	<0.01
NSE_24_	109.0 (7.1–300.0)	39.1 (7.1–210.0)	300.0 (62.0–300.0)	<0.01
NSE_48_	137.5 (9.1–300.0)	33.1 (9.1–290.0)	300.0 (46.6–300.0)	<0.01
NSE_72_	63.7 (10.6–300.0)	20.3 (10.6–300.0)	300.0 (40.2–300.0)	<0.01
CSF S100B, μg/L, median (IQR)				
S100B_i_	3.46 (0.20–30.00)	1.50 (0.20–12.72)	7.88 (2.26–30.00)	<0.01
S100B_24_	4.68 (0.22–30.00)	1.54 (0.22–7.97)	30.00 (1.73–30.00)	<0.01
S100B_48_	3.04 (0.07–30.00)	1.21 (0.07–3.42)	30.00 (2.31–30.00)	<0.01
S100B_72_	1.45 (0.27–30.00)	0.82 (0.27–2.73)	30.00 (1.43–30.00)	<0.01

IQR, interquartile range; CSF, cerebrospinal fluid; NSE, neuron-specific enolase measured immediately (NSE_i_) and at 24 (NSE_24_), 48 (NSE_48_), and 72 h (NSE_72_) after the return of spontaneous circulation; S100B, S100B protein measured immediately (S100B_i_) and at 24 (S100B_24_), 48 (S100B_48_), and 72 h (S100B_72_) after the return of spontaneous circulation.

**Table 3 jcm-10-01531-t003:** Prognostic accuracies of NSE and S100B obtained from CSF for poor neurological outcome.

Values	Cut-off	Sensitivity (95% CI)	Specificity (95% CI)	TP	FP	TN	FN	PPV	NPV
Serum NSE, ng/mL									
NSE_i_ (*n* = 45)	49.3	40.9 (20.7–63.6)	100.0	9	0	23	13	100	63.9
NSE_24_ (*n* = 45)	75.1	40.9 (20.7–63.6)	100.0	9	0	23	13	100	63.9
NSE_48_ (*n* = 45)	54.6	63.6 (40.7–82.8)	100.0	14	0	23	8	100	74.2
NSE_72_ (*n* = 43)	65.3	55.0 (31.5–76.9)	100.0	11	0	23	9	100	71.9
CSF NSE, ng/mL									
NSE_i_ (*n* = 45)	60.3	66.6 (40.7–82.8)	100.0	14	0	23	8	100	74.2
NSE_24_ (*n* = 41)	210.0	80.0 (56.3–94.3)	100.0	16	0	21	4	100	84.0
NSE_48_ (*n* = 40)	290.0	84.2 (60.4–96.6)	100.0	16	0	21	3	100	87.5
NSE_72_ (*n* = 38)	283.0	88.2 (63.6–98.5)	95.0 (75.1–99.9)	15	2	19	2	82.4	94.7
CSF S100B, μg/L									
S100B_i_ (*n* = 45)	12.72	45.5 (24.4–67.8)	100.0	10	0	23	12	100	65.7
S100B_24_ (*n* = 41)	7.97	80.0 (56.3–94.3)	100.0	16	0	21	4	100	84.0
S100B_48_ (*n* = 40)	3.42	94.7 (74.0–99.9)	100.0	18	0	21	1	100	95.5
S100B_72_ (*n* = 38)	2.73	94.1 (71.3–99.9)	100.0	16	0	21	1	100	95.5

CI, confidence interval; TP, true positive; FP false positive; TN, true negative; FN, false negative; PPV, positive predict value; NPV, negative predict value; NSE, neuron-specific enolase measured immediately (NSE_i_) and at 24 (NSE_24_), 48 (NSE_48_), and 72 h (NSE_72_) after the return of spontaneous circulation; CSF, cerebrospinal fluid; S100B, S100B protein measured immediately (S100B_i_) and at 24 (S100B_24_), 48 (S100B_48_), and 72 h (S100B_72_) after the return of spontaneous circulation.

## Data Availability

The data presented in this study are available on request from the corresponding author. The data are not publicly available due to privacy or ethical restriction.

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
