# Peer review of "Comparison of Prognostic Performance between Neuron-Specific Enolase and S100 Calcium-Binding Protein B Obtained from the Cerebrospinal Fluid of Out-of-Hospital Cardiac Arrest Survivors Who Underwent Targeted Temperature Management"

_jcm, 2021, doi:10.3390/jcm10071531_

Round 1

Reviewer 1 Report

In this manuscript, the authors compared the prognostic value of cerebrospinal fluid (CSF) neuron-specific enolase (NSE) and S100 calcium-binding protein B (S100B) on outcomes (survival, favorable neurologic outcomes) in out-of-hospital cardiac arrest (OHCA). They reported that high CSF NSE and S100B with in the first 72 hours of cardiac arrest are strong predictive markers of poor neurological outcome in OHCA survivors. In this univariate analysis the CSF NSE and S100B has good negative predicative value at 48 and 72 hours post ROSC but not at an earlier time points (within 24 hrs). This might be due to low sample size as well as late rise in the CSF NSE and S100B. What are the blood NSF, S100B levels in these patients, at the same time points as CSF values? Can authors perform the multivariable analysis to look for independent variables of neurologic outcomes in this population?

Author Response

Dear. Reviewer

I hope you are doing well in this pandemic era.

Best,

Jung Soo Park

Reviewer 2 Report

General:

This is an interesting study investigating the prognostic value of NSE and S100B levels in CSF. The paper is well written and the results are presented clearly.

Although the authors refer to one of their previous studie in which serum and CSF leveles were compared, the study would have had a higher impact if serum and CSF levels would a have been analyzed also in this cohort of patients.The authors should explain why they did not measure serum levels in parallel.

Another problem affecting the test statistics is that in may patients especially with poor outcome the measured values for NSE and and S100B exceded the upper range of the assays.The authors should comment whether the assay could be adapted. Has the real range of CSF-levels been analyzed at least in a subset of the probes?

However, with this in mind, the study is a good foundation for further studies which might establish e.g. S100B in CSF after 24 or 48 h as a useful marker  for neurological prognostication after cardiac arrest.

Annotations:

  • Table 1: What does the p-value indicate? What is the null hypothesis?.
  • Line 249, 253: Not clear, if S100b in Serum or CSF

Author Response

Dear Reviewer

I hope you are doing well in this pandemic era.

Best,

Jung Soo Park

Round 2

Reviewer 1 Report

Authors have appropriately revised the manuscript. I have no additional queries. I recommend accept as is.

Author Response

We would like to thank the reviewer for evaluating our manuscript. We are really glad that we have met the reviewer’s expectations and addressed the comments appropriately.